# Pediatric Tuberculosis: The Impact of “Omics” on Diagnostics Development

**DOI:** 10.3390/ijms21196979

**Published:** 2020-09-23

**Authors:** Shailja Jakhar, Alexis A. Bitzer, Loreen R. Stromberg, Harshini Mukundan

**Affiliations:** Physical Chemistry and Applied Spectroscopy, Chemistry Division, Los Alamos National Laboratory, Los Alamos, NM 87545, USA; sjakhar@lanl.gov (S.J.); abitzer@lanl.gov (A.A.B.); loreen@lanl.gov (L.R.S.)

**Keywords:** pediatric tuberculosis, diagnostics, omics, biomarkers, lipoarabinomannan

## Abstract

Tuberculosis (TB) is a major public health concern for all ages. However, the disease presents a larger challenge in pediatric populations, partially owing to the lack of reliable diagnostic standards for the early identification of infection. Currently, there are no biomarkers that have been clinically validated for use in pediatric TB diagnosis. Identification and validation of biomarkers could provide critical information on prognosis of disease, and response to treatment. In this review, we discuss how the “omics” approach has influenced biomarker discovery and the advancement of a next generation rapid point-of-care diagnostic for TB, with special emphasis on pediatric disease. Limitations of current published studies and the barriers to their implementation into the field will be thoroughly reviewed within this article in hopes of highlighting future avenues and needs for combating the problem of pediatric tuberculosis.

## 1. Introduction

Tuberculosis (TB) is one of the most common infectious diseases worldwide and continues to pose a substantial threat to pediatric health [1]. According to the World Health Organization (WHO), roughly 10 million individuals were infected with TB in 2019, which resulted in ~1.2 million deaths. Children (<15 years) account for approximately 14% of all TB deaths, and 11% of all TB cases. Of these cases, only 35% of all pediatric TB cases are accurately diagnosed, leading to a delay or lack of treatment [2]. Within this pediatric population, 69% of cases in children under the age of 5, and 40% of cases in children 5–14 years of age remain unreported and undiagnosed, partially accounting for the high infection and mortality rates in this population [3]. Thus, the availability of reliable empirical diagnostics will greatly facilitate improved treatment and survival in children with pediatric TB. However, such diagnostics are currently nonexistent for pediatric TB infection. 

A key contributing factor in our inability to effectively diagnose and treat pediatric TB, is the continued lack of understanding of host-pathogen interactions and disease manifestation in this population. The exact immune mechanisms of underlying TB disease in children are unclear, but some pathways have been elucidated. TB is caused by the bacterium *Mycobacterium tuberculosis* (MTB). In pulmonary manifestation of the disease, the bacteria enter the body via inhalation and colonize terminal alveoli of the lungs after crossing many physical barriers [4]. MTB then activates the host immune response, causing macrophages and lymphocytes to migrate to the infection site. Here, the immune cells begin granuloma formation where MTB can persist in a latent stage for an extended time. Changes in host immune status can cause latent infection to become active at any time [5]. Thus, it is the dynamic balance between bacterial pathogenicity and the host immune system that determines the clinical presentation of TB disease. This balance is influenced by several factors including the infectious dose, virulence and persistence of the pathogen, host health and co-morbidities (HIV/AIDS, diabetes, and others), and the interplay between the innate and acquired immune system [6,7,8,9,10]. 

According to the WHO, successful diagnosis and prompt treatment of TB could prevent millions of deaths each year [2]. However, pediatric TB is not effectively diagnosed by strategies developed for adult infection. Less than 15% of pediatric cases are sputum smear positive, and only 30%–40% of all cases are confirmed by culture [11]. The high failure rate of existing diagnostic tests in pediatrics is largely due to the differential presentation of disease in this population. 

For one, the incomplete maturation of the immune system in pediatrics has been shown to be a contributing factor in disease manifestation and progression [12]. Children aged 1–2 years present with a 20%–30% risk of disease activation, whereas the risk decreases to 5% between ages 3–5, and can potentially further decrease to 2% between 5–10 years of age [13]. Additionally, the developing immune system of children can result in a varied response depending on the stage of disease manifestation, which consequently leads to increased risk of active TB with different disease outcomes [14]. Secondly, childhood TB is often disseminated making it harder to detect via traditional sputum-based diagnostics [15]. Additionally, young children are often unable to expectorate sputum, making the reliance on sputum-based diagnostics difficult for this population [16]. Moreover, pediatric clinical isolates contain fewer bacteria (paucibacillary), making culture and isolation even more challenging. These factors contribute to the challenge of diagnosis of pediatric TB, and render adult diagnostic tests ineffective when applied to children. For instance, while bacterial culture from blood or sputum sample from a presumptive positive patient is the current gold standard TB diagnostic in adults, the method has reportedly low sensitivity when used in children [17]. As a result of the above mentioned factors, a reliable diagnostic for pediatric TB has proved challenging and remains an elusive goal [18,19,20,21,22]. The following section provides a comprehensive assessment of current diagnostic methods with Figure 1 providing a comparison of current approaches for diagnosis of pediatric TB with an “omics” future.

## 2. Current Diagnostics for Pediatric TB

TB can manifest in the latent form in many individuals, and only activates in some patients based on various influencing conditions. Depending on the design, approach, and sensitivity of an assay, a method can be approved for either diagnosis of latent or active forms of the disease in pediatric or adult populations. For instance, the tuberculin skin test (TST) has historically been used worldwide and is currently recommended by WHO for diagnosis of latent TB infection in adult and pediatric populations [2]. In this test format, the tuberculin/purified protein derivative is injected intradermally and the diameter of the inflammatory response on skin is measured. A diameter of greater than 10 mm within two days is considered positive for TB exposure [23]. TST cannot discriminate between latent and active infection, requires two visits to clinic, and is based on subjective interpretation of the spot size [24]. Despite approval for use in children, TST suffers from lower sensitivity in this population, along with poor specificity in individuals exposed to non-tuberculous Mycobacteria or prior bacille Calmette-Guerin BCG vaccination [25]. 

Another test that has been endorsed by WHO for diagnosis of latent TB infection is the interferon gamma release assay (IGRA). This method is based on the quantitative measurement of interferon gamma (IFN γ) released upon activation of innate immune receptors when exposed to MTB antigens, in whole blood. The MTB antigens used includes culture filtrate protein 10, early secretory antigenic target 6, and proteins encoded by genes within the region of difference 1 of the MTB genome [26]. The sensitivity and specificity of IGRA is better than TST, and can differentiate between BCG vaccination and MTB exposure [27]. The assay concept has been commercialized by multiple companies, and available IGRAs include QuantiFERON-TB Gold (QFT-G), QuantiFERON-TB Gold in-tube (QFT-G-IT), and T-SPOT TB (T-SPOT), which vary in the mode of detection (such as lateral flow assays, enzyme-linked immunosorbent assays (ELISA), and ELISpot). The need to draw blood and immediately perform the test is a limiting factor, especially in resource limited areas. 

An inexpensive and simple method commonly used to diagnose active pulmonary TB in low and middle income countries is the sputum smear, which uses microscopic identification of stained MTB in infected samples [28,29,30,31]. WHO has endorsed both light microscopy-based and light emitting diode microscope-based formats of this assay modality for use in both adult and pediatric TB patients. This technique is simple, rapid, and inexpensive, with moderate sensitivity in adults with pulmonary TB [28,29,30,31,32]. However, a drawback of the method is that two sputum specimens are necessary, which are difficult to obtain in all patients, but especially from children [33]. Thus, the value of sputum microscopy for effective diagnosis of pediatric infections is very limited.

Culture is the gold standard for the diagnosis of active TB infection. However, the technique is time-consuming, due to the slow growth rate of MTB. There are two culture-based diagnostic systems approved by the WHO: (1) the liquid culture system with rapid speciation, and culture-based phenotypic drug sensitivity testing (DST) using highly specialized culture media and (2) the mycobacterial growth indicator tube (MGIT). Of the two, MGIT provides higher yield of MTB and significantly faster diagnosis when compared to conventional solid culture, but has the disadvantage of a high cost [34]. However, the paucibacillary and disseminated nature of pediatric TB results in reduced reliability of culture as a gold standard in children (39). Because of these pediatric TB disease states, culture cannot be used to exclude the disease when negative, but is definitely confirmatory when positive. The choice of sample, and the concentration of bacteria for growth are critical considerations in the use of culture as a confirmatory diagnostic. Thus, while culture is considered the gold standard for TB diagnosis in adults, the technique often produces unreliable results in detecting childhood TB [18,19].

The Gene-Xpert MTB/resistance to rifampicin (RIF) assay (Cepheid^®^ USA) was developed to detect DNA sequences specific to MTB, using polymerase chain reaction (PCR). This test has been recommended by the WHO to improve adult and pediatric case detection and identification, and can provide results within 2 h of sample collection [34,35]. Current policy recommends it be used as an initial diagnostic test in children suspected of having multi-drug resistant TB or HIV-associated TB [2]. However, the sensitivity of this assay suffers when bacterial burden is low in samples, as is common in pediatric and HIV positive populations [36]. Furthermore, the assay relies on sputum as a sample, which is a limitation in children, given their inability to expectorate.

Detection of the biomarker lipoarabinomannan (LAM) is a highly promising strategy for pediatric TB because of the non-reliance on sputum as the diagnostic sample [37]. As a result, the MTB cell wall antigen-LAM has gained attention over time. WHO has recommended the use of the lateral flow urine LAM (LF-LAM) assay (Determine^TM^ TB LAM Ag, Abbott) for detection of active TB in severe HIV positive cases. LF-LAM assay involves application of a 60 μL unprocessed urine sample on the test device and results are read visually within 30 min [38]. Another commonly used method to detect LAM in urine are immunoassays, such as ELISA. Here, the capture antibody is used in a multi-well plate, followed by addition of sample and a detection antibody [39,40,41]. However, LAM detection is not yet approved for use in diagnosis of HIV negative pediatric TB, likely because of the lower sensitivity of current diagnostic strategies. Researchers are working on the evaluation of the use of ultra-sensitive sensors in order to circumvent this problem [42,43,44,45,46,47]. The use of samples such as urine and blood favors application of this approach to children, and individuals with disseminated infection. 

While there are promising developments, there are currently no reliable diagnostics for pediatric TB. The percentage of children with active TB that were incorrectly diagnosed by current diagnostic tests, as outlined above, are 40% by culture, 77% by microscopy, and 50% by gene Xpert [48]. The poor reliability of current pediatric diagnostics have caused the diagnosis of childhood TB to be based almost entirely on medical history, clinical symptoms, TST results, and chest radiography [49], and the WHO has a prescribed process for syndromic diagnostics. Therefore, there is an urgent need for development of diagnostics using non-sputum based, reliable biomarkers for detection of tuberculosis in children [44].

Approaches targeting diagnostics development for pediatric TB can broadly be classified into three major categories: (1) detection of antigens or other biomarkers produced by the pathogen, (2) measurement of host immune response to MTB antigens (both humoral and cellular), and (3) unbiased “omics” approaches utilizing genomics, transcriptomics, proteomics, lipidomics, and metabolomics characterization. The use of pathogen signatures and host biomarkers for TB diagnostics have been broadly discussed elsewhere, and is briefly summarized above. This manuscript explores the third approach, using “omics”, for diagnosis of pediatric TB and how this can be applied to pediatric TB diagnosis.

## 3. The Role of “Omics” in TB Diagnostic Development 

Empirical diagnostics can potentially traverse the challenges associated with various presentations and manifestations of TB disease in children, as well as address the varied manifestations of TB disease in adults (disseminated, extra-pulmonary, drug-resistant, and latent) [50]. Molecular signatures to facilitate such diagnostics can belong to any of the “omic” categories of relevance and the use of omics as a tool for biomarker discovery has advanced greatly over the past decade, facilitating such development. As depicted in Figure 2, a well-rounded omics approach for investigating TB pathology includes genomics, transcriptomics, proteomics, metabolomics, and lipidomics. Such a high throughput approach can provide researchers with an unbiased multi-dimensional understanding of disease progression and outcomes, to better develop an all-encompassing diagnostic test.

In comparison to current diagnostics, the average response times for the various omics technologies discussed in this manuscript vary greatly at this point in development. For instance, a technically naive health care worker can accomplish running an Alere LAM assay at the point of need, within 30 min for a low cost. However, running a genomics panel on patient sputum samples is far more expensive (hundreds of dollars, depending on the method), and may require pathogen concentration via culture. This is technically intensive and time consuming and requires extensive bioinformatics capabilities, which in turn require skilled capabilities and complex laboratory infrastructure. Whereas, the Alere immunoassay interrogates for one single biomarker of interest, genomic arrays can provide a pan-diagnostic approach for discriminative diagnosis of infection. Further, genomic technologies are advancing with respect to ease of use and operation at unprecedented rates, and, in fact, have defied Moore’s law [51]. Field forward sequencing capabilities that can be used quickly and in an automated version at the point of need are rapidly emerging [52]. Many culture-free sequencing capabilities are also emerging, decreasing the time to result, especially when combined with deployable and easy to use informatics pipelines [53]. Similar to genomics, proteomic and metabolomic arrays are also time consuming, labor intensive, and expensive in their current form. However, this is also changing rapidly, albeit not as quickly as evidenced with genomics. Service centers providing proteomic microarray development and validation are emerging, and despite the research required and custom development involved, are now available for ~$100 per sample [54]. Thus, all of these omics-based methods hold more promise for the future, because of their holistic nature, flexibility, and agility to being applied to a variety of human health challenges. These properties are especially relevant to pediatric TB, where the reliability of current diagnostics is very poor. A novel omics future promises a superior, more reliable strategy for pediatric TB diagnosis. Currently, there are various researchers working on the development of deployable and easy to use informatics pipelines for proteomics, genomics, and other omics strategies [55,56,57,58].

An omics approach would expand our knowledge of diagnostic biomarkers and could also facilitate better understanding of MTB pathogenesis and drug resistance mechanisms to aid in the development of suitable therapeutics and vaccine candidates. An example of this would be using proteomics to measure cellular activity to provide deeper insight into pathogen or host cellular processes during different stages of infection [59]. Additionally, whole genome sequencing (WGS) can be used to identify signatures of drug resistance in the pathogen before or during drug therapy to provide a customized pharmaceutical regimen to improve treatment efficacy [60]. With the evolution of machine learning and artificial intelligence based computational capabilities, an “omics” approach can be utilized in many ways to identify host or pathogen signature patterns for diagnostics and targets for therapeutics.

### 3.1. Genomics

WGS of MTB presents an exciting opportunity with respect to improved strategies for diagnosis of TB, irrespective of disease state. Recent advances in genomics have allowed for the use of WGS in order to discriminate between reinfection versus relapse of TB infection. Unlike serological techniques, WGS as a diagnostic tool can confirm the presence of current MTB infection [79]. While most WGS-based work has required prior culture of the pathogen, there have been two recent studies using WGS as a confirmatory diagnostic test by sequencing MTB genomes directly from uncultured sputum samples [80]. This is of great relevance in the progress towards using this technique as a diagnostic strategy. However, it is important to note that MTB DNA was also detected in samples that were culture and smear negative. Therefore, DNA-based genomic detection cannot distinguish between active and cleared infections where residual DNA may be present from dead bacteria [61].

One critical advantage of WGS-based diagnosis of TB infection is the ability to identify, and appropriately treat mixed MTB infections (up to 50% in certain TB endemic regions), which are defined as disease caused by more than one distinct MTB strain [80,81]. Existing platforms have consistently demonstrated lower sensitivity in mixed infections, suggesting that WGS may fill a major gap in this arena. For instance, the Xpert assay for rifampicin resistance on mixed infections has a lower sensitivity of (80%), compared to 93% on homogenous infections [81]. A second advantage of WGS is the data on its application towards the determination of drug resistance. Sequencing data consistently agrees with conventional DST, and is associated with shorter turnaround times, especially when done from early cultures [60]. The same reasons that make WGS suitable for tackling these challenging problems, also make it an excellent candidate for tracking multiple TB presentations in pediatric populations. However, most often, WGS requires culture prior to sequencing, which can delay diagnosis, and complicate application in paucibacillary pediatric cases. More studies such as the ones noted above [54] need to be performed on non-sputum samples to explore culture-free WGS strategies. In addition to diagnostics, DNA microarray chips have been explored for the rapid detection of MTB resistance to different therapeutics. One such study collected sputum samples from 42 patients with TB and determined a 92.8% susceptibility and 93.8% specificity for the identification of resistance to the antimycobacterial drug, rifampicin [82]. A similar study assessed 176 clinical isolates on an array with 12 pairs of primers, and 60 nucleotide polymorphisms of 9 different MTB genes, and compared the results to culture-based DSTs, GenoType MTBDR*plus*, and MTBDR*sl* tests. It was found that the array was able to detect for resistance to isoniazid with a sensitivity of 100% and a specificity of 96.7%, whereas for rifampicin it was observed to be 99.4% and 96.7%, respectively. These outcomes present excellent suitability for reliable use in a clinical setting for the identification and monitoring of resistant strains. Development of such methods could have a large impact not only on diagnosis of MTB, but also on disease prognosis [83].

Aside from diagnostics, genomics has proved to be a powerful tool for understanding the molecular epidemiology of TB, mechanism of drug resistance, and for unraveling protein and structural information. Analysis of WGS information is also useful for epidemiological characterization and tracing transmission [79,84,85], and the genetic sequence data provides a code for protein structure. The TB Structural Genomics Consortium is an organization dedicated to determining protein structures for MTB proteins on a genome wide scale. Large scale structural determination could help researchers design smarter drugs to combat disease [62].

### 3.2. Transcriptomics 

Compared to genomics, transcriptomics is more readily adaptable for rapid diagnostic development. Analysis of host coding RNA can be used to investigate gene expression patterns throughout the course of the disease, and potentially with varied manifestations, making them ideal targets for investigation of pediatric TB diagnostics. Non-coding RNA, which does not encode for any protein, is frequently linked to regulatory functions, which may be altered during different disease states. A recent study identified 15 non-coding micro RNA (miRNA) as a signature for TB disease [63]. In this study, researchers compared the miRNA profiles from different genetic backgrounds, adult patients with active pulmonary and extra-pulmonary infections, TB/HIV co-infections, and latent TB. Using miRNA as a marker was found to have an overall sensitivity of 86% and a 79% specificity for diagnosis of active TB infection in these varied populations. Using RNA sequencing of whole blood samples, researchers examined the use of small non-coding RNA population such as miRNA, PIWI-interacting RNA (piRNA), small nucleolar RNA (snoRNA), and small nuclear RNA (snRNA) as host biomarkers for active and latent MTB in a systematic manner [86,87]. From this approach, one miRNA and two piRNAs were identified as potential biomarkers for latent MTB [88], but further studies are required for their validation in such an application.

There have also been several transcriptomic studies in pediatric populations with TB in the last 10 years [64,65,66,89,90,91,92]. However, the broader implications of these results towards the development of a pediatric diagnostic are limited by the lack of diversity in population sampling. Host circulating miRNA profiles from whole blood in a pediatric TB population, were analyzed by Zhou et al. who demonstrated that a combination of eight miRNA signatures provided a 95.8% sensitivity and 100% specificity for the discrimination of children infected with TB versus uninfected healthy controls [65]. Using the GeneChip Human Exon 1.0 ST Arrays (Affymetrix), a 116 gene signature set from whole blood was identified in 27 Warao Amerindian children (9 active TB, 9 latent TB, and 9 healthy controls) [64], again suggesting that a transcriptomic profile assessment can provide a reliable strategy for diagnosis of pediatric TB. The researchers further validated the ten genes in an independent cohort of 54 children by using quantitative real time polymerase chain reaction (qRT-PCR), and found that five out of ten genes were sufficient to achieve 78% sensitivity and 100% specificity in this population [93]. Additionally, a recent study with Indian children explored transcriptomic profiles from peripheral blood in various stages of disease presentation, and identified 12 transcriptional immune biomarkers that could differentiate between infected and asymptomatic children [66]. Whereas a comprehensive and well-characterized pathogen transcriptomic study is required in order to benchmark relevant signatures, the early developments in this field show promise for the application of these signatures for pediatric TB diagnosis.

A modified approach to transcriptomic diagnosis is to track changes in host cells, rather than the pathogen. For example, by focusing on transcripts from whole blood samples of patients infected with TB, researchers identified a profile of 86 host transcripts which can potentially distinguish TB infections from others, and an additional 393 which further characterize the infection as active or latent [67]. Similar studies in pediatric cohorts would help further understanding of TB-host interaction, and also facilitate differentiation of latent and active disease, thereby advancing diagnostic applications. Transcriptional signatures are independent of bacterial load, which strengthens the argument for using this technique for diagnosing low burden cases such as in paucibacillary disease, as evidenced in children. 

### 3.3. Proteomics 

Proteomics profiling has been used to measure cellular activity, and can provide a deep insight into cellular processes in complex clinical backgrounds. Understanding the wide array of proteins expressed by both MTB and the host in response to MTB infection could shed light on pathways responsible for pathogenesis and persistence [94,95]. Such proteomic studies could target the pathogen-specific proteome, or host signatures in response to MTB infection, both of which have been attempted extensively, and examples from which are discussed below [96,97]. 

The function of about one-quarter of the MTB coding genome and the precise activity and protein networks of most of the associated proteins remain poorly understood. Protein mass spectrometry and functional proteomics have provided new insights into making this information more accessible to diagnostics development. Early proteomic studies used two-dimensional gel electrophoresis (2D-GE) to analyze proteins from bacterial fractions and culture supernatants of MTB [59]. However, the low resolution of this method limited the analysis to only a few hundred proteins [98,99,100,101], which is insufficient to provide a clear assessment of the signature array. The use of liquid chromatography-tandem mass spectrometry (LC-MS/MS) shotgun proteomic methods in both targeted and non-targeted studies has allowed for the expansion of this capability to several thousand proteins at a given time [68,102]. More advanced MS techniques, such as selected reaction monitoring, have allowed for the quantification of ~80% of the MTB proteome, and do not require cell fractionation or separation [68]. In addition to MS, proteome microarrays have also been used to profile thousands of protein interactions in a single experiment [103,104]. Proteomic arrays have enabled researchers to define an immunoproteome for MTB. Until recently much of biomarker discovery has relied on traditional methods for separation and identification, however, as alternative methods are being constantly described the field has grown. To date there are three proteome-wide screening approaches that have been employed for the identification of candidate antigens for CD4+ T cell responses to MTB. All three studies found that a relatively small percentage of the proteome was responsible for the majority of the immune response [97,105,106].

In addition to an immunoproteome, a proteomic microarray approach has been used to screen 4262 MTB antigens from 40 adult TB patients which allowed for the identification of 152 MTB antigens that were differentially elevated among patients with active versus latent disease [69]. Yet another study used a two-way proteome microarray approach to screen 84 potential host MTB interactors in infected adults, developing a signature repository that can be further used to understand MTB pathogenesis [70]. Deng et al. identified 14 adult serum biomarkers to differentiate between patients with active disease and those that have recovered from TB infection, facilitating monitoring of treatment outcomes [71]. In addition to presenting the proteome library, the investigators were also able to begin to explore the use of such microarrays in determining protein–protein interactions, biomarker discovery, and differentiating between individuals with active disease and those that had recovered from TB infection, demonstrating the potential usefulness of such platforms for real-world applications.

From a longitudinal cohort of 6,363 MTB positive, HIV-negative adolescents of ages between 12–18 years in South Africa, host protein signatures associated with MTB were systematically assessed. In this study, the cohort was followed for 2 years and investigators reported that 46 individuals developed microbiologically confirmed MTB disease, while 106 non-progressors were identified. As such, 3000 human host proteins from plasma were quantified, of which 361 were found to demonstrate significant difference in abundance between individuals with microbiologically confirmed TB and non-progressors. From these 361 proteins, a 5-protein signature, TB risk model 5 (TRM5), was further sub-selected for use in discriminatory diagnostics. A second 3-protein pair (3PR) was further added to this sub-selection in order to improve the efficacy of the diagnostic platform. However, neither the TRM5 or 3PR achieved the minimum criteria for an incipient TB test as defined by the Foundation for Innovative New Diagnostics (FIND) or WHO, and, therefore, additional work is still needed to improve these signature-based protein assays [107]. Additionally, a subset of proteins that are exported, termed the exportome, could be potential source of additional disease biomarkers. Efforts to identify exported proteins have been typically limited to in vitro work. However, recently an in vivo method has been described and termed EXIT (exported in vivo technology) for the discovery of MTB exported proteins, as demonstrated in murine infection models. Over 500 proteins were revealed to be exported, several of which were induced in vivo. Proteins discovered by this technique should be further explored as potential biomarkers for adult and pediatric MTB disease markers [108].

In a third study, researchers identified an eight-protein host signature which had ramifications for the diagnosis of TB disease. In this study, three separate cohorts were enrolled for a total of 640 individuals. The initial cohort of individuals was used for the screening of protein biomarkers of TB, the second to establish and test the predicted model, and the third for biomarker validation. The initial round of screening involved a microarray comprised of 16 non-overlapping arrays to measure 640 human proteins. Sixteen proteins of interest were then further analyzed in a second array. Using a series of mathematical models, a diagnostic model was built using an eight-protein signature. In the second test cohort the signature had an 83% specificity and a 76% sensitivity. The third cohort, in which the signature was validated, the specificity and sensitivity was 84% and 75%, respectively. While this study was done with adults, a similar study could be designed for pediatric MTB to develop a pediatric specific model [109].

Proteomic profiles can allow for the diagnosis of pediatric TB, and despite the disease’s varied manifestations, several researchers have begun to specifically explore that possibility. For instance, a quantitative proteomics approach using LC-MS/MS was employed to characterize plasma from 72 children in different test groups (active TB, inflammatory disease control, and healthy control) at a Beijing Children’s Hospital. The study identified 49 proteins in pediatric cases that were differentially expressed between active and latent TB [72]. One study characterized the plasma proteins in children at different MTB infection stages (active TB and LTBI), and identified four proteins—XRCC4, PCF11, SEMA4A, and ATP11A—to be signatures of active TB disease using proteomics [72]. Given the differential presentation of pediatric TB disease, it is likely that a combinatorial approach exploring varied biomarker signature profiles may provide a greater reliability of identification rather than a single factor approach [110].

### 3.4. Lipidomics 

Lipids are an essential player in biological processes, both within the host and the pathogen. MTB is known to have one of the most complex lipid envelopes in nature, which forms the barrier between the pathogen and the host, and a substantial lipid biosynthesis capacity within their genome [111]. The multilayered cell wall contains both an inner phospholipid bilayer as well as an outer lipid layer consisting of mycolic acid. Lipids from both layers have strong potential to be used as a biomarker for diagnosis.

With considerable advances in mass spectrometry over the past decade, the field of lipidomics has advanced MTB research. The large-scale characterization and quantification of lipids has led to the development of MycoMass, a database for mycobacterial lipids [73]. The database currently contains 58 lipid types. More than 40 of these mycobacterial lipids lack any similarity to other eukaryotic or Gram-negative organisms, making them unique signatures for diagnostics development and therapeutic targeting. The unique Mycobacterial lipidome is, thus, a signature repository that requires further characterization [112]. When comparing the lipid profile from normally grown MTB with dormant and reactivated bacteria, analysis revealed a total of 4187 significant features with 2480 features found to have significant variation during the transition from normoxial growth to dormancy. Across the three different stages a total of 74 fatty acyls showed significant variations. Findings such as these could play roles in the discovery of biomarkers for various different stages of MTB [113].

Mycolic acids (MAs) are a main component of the cell wall of mycobacteria. The cell wall of mycobacterium provides protection against a host immune response by mediating macrophage trafficking events, and by helping the bacterium to grow within host macrophages [114,115,116]. MTB has three different structural classes of MAs namely, alpha-, methoxy-, and keto-MAs [117], with the most abundant form being α-MA (>70%), and methoxy- and keto-MAs being minor components (10% to 15%) [118]. MA-classes play a crucial role in virulence and are present in high concentrations in the bacterial cell wall. While they are unique to Mycobacteria, these lipids differ considerably between different Mycobacterial species/strains, a factor which could be helpful for differential diagnosis. For example, the detection of all three forms of MAs in bacterial extracts and gamma irradiated whole bacteria, using surface enhanced Raman spectroscopy [119]. While some studies have used a surface plasmon resonance (SPR) technique to detect serum antibodies to MAs, the method has not been validated clinically [120]. Using a biosensor platform to detect antibodies against MAs from patient serum improved MA detection (sensitivity 91.3% in TB and HIV positive patients) compared to ELISA [121]. Other techniques used to detect MAs are LC-MS and high-performance liquid chromatography, which are both expensive and require advanced user training to operate and interpret data, making them difficult to use in resource-poor settings [119,122]. MAs were detected in adult TB sputum by Shui et al. The study demonstrated a sensitivity of 94% and specificity of 93% in discriminating between TB cases and controls [74]. The authors speculated that the method “might offer advantages in specialized situations such as pediatric cases (where sputum volume is very limited)”, although clinical validation of this has yet to be performed. Lipidomic and metabolomic analysis of MAs in samples such as urine and serum in pediatric and adult patients is needed to establish the effectiveness of this lipidic signature as a potential diagnostic biomarker [123].

Other prominent mycobacterial cell wall components include lipoglycans such as trehalose dimycolate (TDM), phosphatidyl-myo-inositol mannosides (PIM), LAM, and lipomannan (LM) [124,125,126]. Animal studies have shown that lipids on mycobacterial surface interfere in their interaction with phagocytes, thereby influencing pathogenesis [127]. However, not much is known about the molecular mechanism with exception of key lipoglycans, LAM and LM [127,128].

LAM is an amphipathic molecule released from metabolically active or degrading bacterial cells resulting in the activation of host immune response [129,130,131]. In 2001, Hamasur et al. discovered that LAM was detectable in the urine several hours after intra-peritoneal injection of crude MTB cell wall extract into mice [132]. The observation provided researchers with an opportunity to evaluate LAM as a biomarker for the development of non-invasive [132,133] point-of-care tests for TB. As a result, a lateral flow urine LAM assay is currently available (Determine™ TB LAM Ag, Abbott Biotechnologies), with a sensitivity of 45% and specificity of 92% in HIV positive patients [134] and is recommended by the WHO only for use in HIV-positive adults with CD4 counts less than or equal to 100 cells·μL^−1^ presenting with symptoms of TB [38]. The guidelines for use of urine LF-LAM assay are similar in children, based on data from adults [38]. Previous work from our team demonstrated the detection of urinary LAM at a maximal concentration of 350 pM in individuals without HIV co-infection using a sandwich immunoassay on an ultra-sensitive waveguide based optical biosensor [42]. The conclusions of this work are supported by a recent study using an improved chemiluminescence readout, with sensitivity and specificity of 93% and 97%, respectively [47]. These findings show that a more sensitive assay format is required in immunocompetent individuals.

Despite advances in LAM diagnostics testing for adults, data on pediatric testing is still scarce. A WHO update on urine LAM assays reported a pooled sensitivity of 47%, and a pooled specificity of 82% among various studies performed in children with HIV [135], which is also reflected in independent assessments of both lateral flow and ELISA formats of detection [16,39,136]. These studies show that LAM measurement is more reliable in immunocompromised children, and that measured concentrations of the antigen decrease with anti-TB treatment [136], which suggests potential for this biomarker to be used as a prognostic indicator. Longitudinal studies demonstrating antigen concentrations as a function of disease progression and treatment must be performed in order to validate this hypothesis, as previously demonstrated for the measurement of lipomannan in *M. bovis* infection [137,138,139,140]. 

Previous work from our group has demonstrated that LAM is associated with high-density lipoproteins (HDL) in host blood [45]. This association should be considered when developing diagnostics, as with membrane insertion and lipoprotein capture methodologies [43,137,141,142,143,144], because traditional strategies for measuring the monomeric antigen are likely to be unsuccessful in this conformation. To date there are only a few studies showing detection of LAM in blood from adults [46,47,133] and none in a pediatric population.

Lipidic profiles of mycobacteria are unique—and targeting these differential biomarkers can definitely provide a unique strategy for the diagnosis of active disease. It is important to develop capabilities for the characterization and measurement of pathogen and host lipids, and further improve our understanding of pathogen lipid profiles in order to advance this field of science. Furthermore, the genes and enzymes that regulate these specific lipids can be identified and pave a way to integrate other omics approaches, such as metabolomics. 

### 3.5. Metabolomics

Metabolomics can be used to study changes in host metabolism and associated processes in response to TB infection. A recent study in children from the United Kingdom and Gambia demonstrated alterations in host metabolism using ^1^H NMR spectroscopy and MS, to provide a signature repertoire for diagnostic applications. ^1^H NMR data was analyzed to discriminate between children with TB, and those infected with other diseases, and demonstrated a sensitivity and specificity of 69% and 83%, respectively, for the diagnosis of TB. MS characterization of metabolic profiles had similar results with a sensitivity and specificity of 67% and 86%, respectively [75]. The study also showed raised levels of ceramide, a type of sphingolipid found in high concentration in cell membranes. Ceramides have been shown to contribute to maturation of the phagosome in macrophages infected with MTB, causing increased killing of pathogenic MTB [145], and can serve as unique metabolic signatures of pathogenesis. Similar alterations in metabolic markers have been seen in adult studies [76,77,78]. Thus, metabolomics studies can provide some useful insights into understanding pediatric immune response mechanisms to MTB. However, large scale and controlled studies are required for the identification of metabolomic processes and signatures that can be used for diagnostic applications. Yet, the change in host metabolome in response to TB infection can likely provide useful information for pediatric TB diagnosis and understanding of pathogenesis.

## 4. Conclusions

Pediatric TB is a devastating problem worldwide with no reliable diagnostic tests to differentiate between latent and active TB cases or help guide treatment to success. Current pediatric diagnostics are typically based on difficult-to-collect sputum samples, are time consuming, and associated with lower sensitivity. Further, current diagnostic tests do not address the varied challenges of differential presentation of TB disease in children when compared to adults. Some of the current approaches can potentially be refined and re-aligned for pediatric applications. For instance, assay modalities with enhanced sensitivity or blood-based detection methods can greatly improve the use of LAM as a diagnostic marker for pediatric TB disease. However, there is a distinctive need for a broader search for empirical signatures, validated strategies, comprehensive and reproducible assessments, and clinical studies that target pediatric presentation of TB. Due to incredible complexity of TB pathology in various populations, using an “omics-based” approach can facilitate the identification of suites of biological signatures necessary for developing a universal TB diagnostic test. While many “omics” studies have been done in adult populations, these results only provide a road map for building further understanding of pediatric TB pathology. Additional investigation of how TB stages differ and progress to active disease in a pediatric population would aid development of an all-encompassing specific and sensitive diagnostic test.

## Figures and Tables

**Figure 1 ijms-21-06979-f001:**
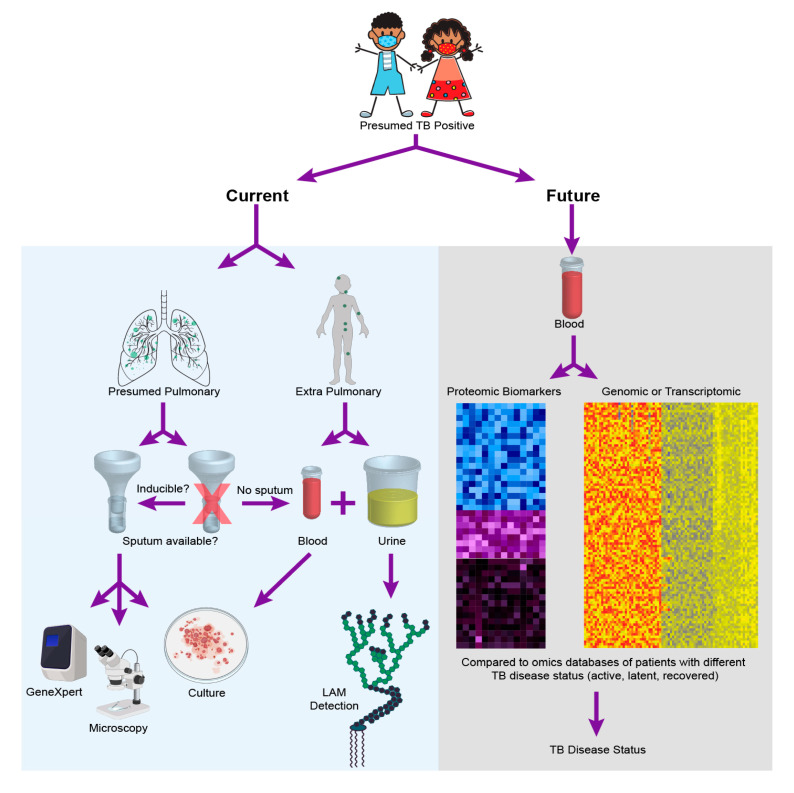
Pediatric tuberculosis (TB) diagnostics—today and tomorrow. The figure outlines the diagnostic choices and decisions that are made when a child is presumed positive for TB disease today, while highlighting that these choices and challenges may be entirely alleviated with the realization of empirical diagnostics as facilitated by one of the many potential omics strategies discussed in this review. To create the cumulative figure, images 379758506 by katy_k20 and 394943974 by janista were obtained from DepositPhotos and used under the standard license agreement. Additional images were downloaded from BioRender.com and used under licensed agreement.

**Figure 2 ijms-21-06979-f002:**
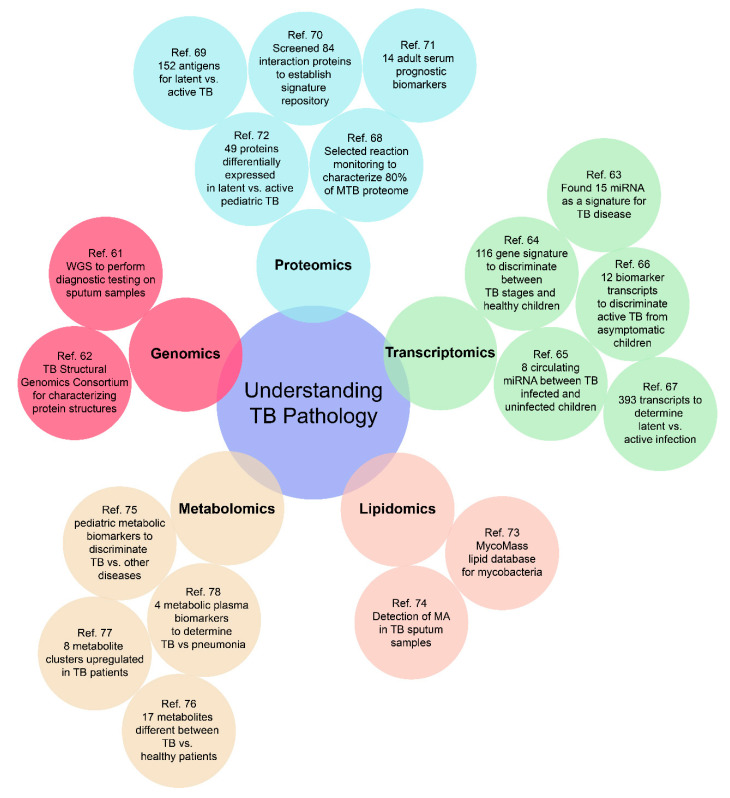
Representative cases for an “omics-based” approach to build a comprehensive understanding of the pathology of both adult and pediatric TB. There are still many omics-based approaches to be further investigated, especially for pediatric TB [61,62,63,64,65,66,67,68,69,70,71,72,73,74,75,76,77,78].

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
