# Peer review of "Pediatric Tuberculosis: The Impact of “Omics” on Diagnostics Development"

_ijms, 2020, doi:10.3390/ijms21196979_

Round 1
Reviewer 1 Report
The paper “Pediatric Tuberculosis: The Impact of “Omics” on Diagnostics Development” describes the omics development in the field of pediatric tuberculosis. The author introduced the current diagnostics for pediatric TB and explained the development of genomic, proteomics, transcriptomics, lipidomic, and metabolomics. The overall review of the field is broad, but some details need justification. I would suggest a significant revision before it was proper for the publication on International Journal of Molecular Sciences.
The suggestions are as follows.
- There are “host” or “pathogen (Mtb)” in all omics level studies. It was not clearly distinguished in the review, which makes it hard to tell while reading.
- The only figure in the manuscript has the information that was more suitable for a table. Meanwhile, a critical schematic figure would be better prepared.
- Just be looking at figure 1, one would expect the main text about the studies of transcriptomics and proteomics, while the most described studies are about the LAM. The emphasis for each omics study would be reasonable, but the distribution should be carefully maintained.
- There is a fair amount of proteomic studies reported, and it will be hard to summarize in two paragraphs. A proper expansion of the studies using proteomics would be required. For instance, in these studies, the host proteomics studies, the antigen proteomics studies, the protein microarray studies were not well described.
- There are at least three kinds of Full Mtb protein array reported. The current review only mentioned one of them.
- LAM would be of great value for diagnostics. A proper truncation and summary would be appreciated.
- Besides the LAM, a summary of other lipoglycans is suggested.
Author Response
Manuscript ID # ijms-902902
Pediatric Tuberculosis: The Impact of “Omics” on Diagnostics Development
Authors’ Response to the Reviewers’ Comments:
We thank the reviewers for a meticulous assessment of our manuscript and making useful comments/remarks for further improvements. Our point-by-point responses to the reviewers’ comments are provided below.
Reviewer 1:
- There are “host” or “pathogen (Mtb)” in all omics level studies. It was not clearly distinguished in the review, which makes it hard to tell while reading.
The manuscript focuses on the interplay between host and pathogen, and how the signatures associated with both can be utilized for improved diagnostics development for pediatric TB. To that end, our discussion of pathogen and host signatures were blurred in the original submission, and we have attempted to clarify this throughout the manuscript, following the reviewer’s comments. Whereas the changes are interspersed throughout the manuscript, some major changes are highlighted as follows: Page 10- Line 192, 194 and 197; Page 11- Line 201; Page 13- Line 246; and Page 14- Line 262.
- The only figure in the manuscript has the information that was more suitable for a table. Meanwhile, a critical schematic figure would be better prepared.
Thank you for this suggestion. A more comprehensive figure that envelops the overall scheme of the manuscript has been developed and has been incorporated as Figure 1, on page 4 of the revised document.
- Just by looking at figure 1, one would expect the main text about the studies of transcriptomics and proteomics, while the most described studies are about the LAM. The emphasis for each omics study would be reasonable, but the distribution should be carefully maintained.
We have revised the manuscript to equally emphasize the different “omics” aspects that are described in Figure 1 more evenly. The specific changes are outlined as below:
On Page 12 and 13, with respect to the Genomics section, the following text has been included: “In addition to diagnostics, DNA microarray chips have been explored for the rapid detection of MTB resistance to different therapeutics. One such study collected sputum samples from 42 patients with TB and determined that 92.8% susceptibility and 93.8% specificity for the identification of resistance to the antimycobacterial drug, rifampicin. A similar study assessed 176 clinical isolates on an array with 12 pairs of primers, and 60 nucleotide polymorphisms of 9 different MTB genes, and compared the results to the culture-based DST, GenoType MTBDRplus and MTBDRsl tests. They found that their array was able to detect for resistance to isoniazid with a sensitivity of 100 %, and a specificity of 96.7%, whereas for rifampicin it was observed to be 99.4%, 96.7%. These outcomes present excellent suitability for reliable use in a clinical setting for the identification and monitoring of resistant strains. Development of such methods could have a large impact not only on diagnosis of MTB, but also on disease prognosis”.
On Page 14, with respect to the transcriptomics section, the following text has been included: “Using RNA sequencing of whole blood samples, researchers examined the use of small noncoding RNA population such as miRNA, PIWI-interacting RNA (piRNA), small nucleolar RNA (snoRNA), and small nuclear RNA (snRNA) as host biomarkers for active and latent MTB in a systematic manner. From this approach, one miRNA and 2 piRNAs were identified as potential biomarkers for latent MTB, and further studies are required for their validation in such an application”.
On Pages 16 and 17, with respect to the Proteomics section, the following text has been included: “From a longitudinal cohort of 6,363 MTB positive, HIV-negative adolescents of ages between 12-18 years in South Africa, host protein signatures associated with MTB were systematically assessed. In this longitudinal study, the cohort was followed for 2 years and investigators reported that 46 individuals developed microbiologically confirmed MTB disease, while 106 non-progressors were identified. 3,000 human host proteins from plasma were quantified , of which 361 were found to demonstrate significant difference in abundance between individuals with microbiologically confirmed TB and non-progressors. From these 361 proteins, a 5-protein signature, TB Risk Model 5 (TRM5), was further sub-selected for use in discriminatory diagnostics. A second 3-protein pair (3PR) was further added to this sub-selection in order to improve the efficacy of the diagnostic platform. However, neither the TRM5 or 3PR have achieved the minimum criteria for an incipient TB test as defined by FIND and WHO, and therefore additional work is still needed to improve these signature-based protein assays [96]. Further studies such as this, and the concomitant development of sensitive diagnostic platforms can transform the landscape of empirical molecular diagnostics”.
- There is a fair amount of proteomic studies reported, and it will be hard to summarize in two paragraphs. A proper expansion of the studies using proteomics would be required. For instance, in these studies, the host proteomics studies, the antigen proteomics studies, the protein microarray studies were not well described.
We have further expanded on the proteomics section, as indicated in the response to question 3, above.
- There are at least three kinds of Full Mtb protein array reported. The current review only mentioned one of them.
We have rectified this oversight. On Page 16 and 17 of the document, the revised proteomics section now includes a description of the three full Mtb protein arrays, as suggested by the reviewer. This section now reads “Proteome microarrays have also been used to profile thousands of protein interactions in a single experiment. Using such a proteomic microarray approach, 4,262 MTB antigens from 40 adult TB patients screened which allowed for the identification of 152 MTB antigens that were differentially elevated among patients with active versus latent disease. Yet another study used a two-way proteome microarray approach to screen 84 potential host MTB interactors in infected adults, developing a signature repository that can be further used to understand MTB pathogenesis. Deng et al. identified 14 adult serum biomarkers to differentiate between patients with active disease and those that have recovered from TB infection, facilitating monitoring of treatment outcomes. From a longitudinal cohort of 6,363 MTB positive, HIV-negative adolescents of ages between 12-18 years in South Africa, host protein signatures associated with MTB were systematically assessed. In this longitudinal study, the cohort was followed for 2 years and investigators reported that 46 individuals developed microbiologically confirmed MTB disease, while 106 non-progressors were identified. 3,000 human host proteins from plasma were quantified , of which 361 were found to demonstrate significant difference in abundance between individuals with microbiologically confirmed TB and non-progressors. From these 361 proteins, a 5-protein signature, TB Risk Model 5 (TRM5), was further sub-selected for use in discriminatory diagnostics. A second 3-protein pair (3PR) was further added to this sub-selection in order to improve the efficacy of the diagnostic platform. However, neither the TRM5 or 3PR have achieved the minimum criteria for an incipient TB test as defined by FIND and WHO, and therefore additional work is still needed to improve these signature-based protein assays. Further studies such as this, and the concomitant development of sensitive diagnostic platforms can transform the landscape of empirical molecular diagnostics”.
- LAM would be of great value for diagnostics. A proper truncation and summary would be appreciated.
A summary has now been included on Page 7 of the revised document, and reads as “Detection of the biomarker Lipoarabinomannan (LAM) is a highly promising strategy for pediatric TB because of the non-reliance on sputum as the diagnostic sample. As a result, the MTB cell wall antigen-LAM has gained attention over time. WHO has recommended the use of the lateral flow urine LAM (LF-LAM) assay (DetermineTM TB LAM Ag, Abbott) for detection of active TB in severe HIV positive cases. LF-LAM assay involves application of 60 μL unprocessed urine sample on the test device and results are read visually within 30 minutes. Another commonly used method to detect LAM in urine are immunoassays, such as ELISA. Here, the capture antibody is used in a multi-well plate, followed by addition of sample and a detection antibody. However, LAM detection is not yet approved for use in diagnosis of HIV negative pediatric TB, likely because of the lower sensitivity of current diagnostic strategies. Researchers are working on the evaluation of the use of ultra-sensitive sensors in order to circumvent this problem. The use of samples such as urine and blood favors application of this approach to children, and individuals with disseminated infection. ”
- Besides the LAM, a summary of other lipoglycans is suggested.
A brief summary about other lipoglycans has been added on page 19 and 22 and now reads “Other prominent Mycobacterial cell wall components include lipoglycans such as trehalose dimycolate (TDM), phosphatidyl-myo-inositol mannosides (PIM), LAM and lipomannan (LM). Animal studies have shown that lipids on Mycobacterial surface interfere in their interaction with phagocytes, thereby influencing pathogenesis. However, not much is known about the molecular mechanism with exception of key lipoglycans, LAM and LM. Present day understanding of LAM/LM structure comes from the pioneering work by Hunter, Chatterjee, Brennan and coworkers, who identified these antigens as amphipathic molecules released from metabolically active or degrading bacterial cells resulting in the activation of host immune response.
Similarly, it follows that LM from M. bovis can potentially be a signature for the identification of bovine tuberculosis. Indeed, our group and others have measured this lipoglycan in infected animals, demonstrating feasibility for such a hypothesis”.
Reviewer 2 Report
Interesting paper because it deals with a topic of extreme importance: the diagnosis of pediatric tuberculosis.
Points to be better clarified:
1) which biological materials the Omics can be used
2) average response times
3) costs in relation to classic methods ( cultures, genexpert..)
Author Response
Manuscript ID # ijms-902902
Pediatric Tuberculosis: The Impact of “Omics” on Diagnostics Development
Authors’ Response to the Reviewers’ Comments:
We thank the reviewers for a meticulous assessment of our manuscript and making useful comments/remarks for further improvements. Our point-by-point responses to the reviewers’ comments are provided below.
Reviewer 2
1) which biological materials the Omics can be used?
While depending on the method employed, omics can be used to interrogate a variety of clinical samples such as blood, urine, sputum and others in the infected host, the unifying aspect of an omics-based approach is that it can potentially allow for the investigation of various presentations of the disease in a single sample, blood. The range of current samples that have been evaluated by each of the omics methods discussed in the manuscript, and the potential for future blood-based diagnostics are now more elaborately discussed throughout the manuscript as follows: Page 12- Line 224, Page 14- Line 263, 266, 273. This concept has also been discussed in the new schematic 1 included with the revised manuscript.
2) average response times- The time to result, 3) costs in relation to classic methods (cultures, genexpert..)
We have attempted to address questions 2 and 3 from the reviewer together, as below. This revised text has been incorporated into the manuscript on page 9 and 10 as well, and now reads as “In comparison to current diagnostics, the average response times for the various omics technologies discussed in this manuscript vary greatly at this point in development. For instance, a technically naive health care worker can accomplish running an Alere LAM assay at the point of need, within 30 min for a low cost. However, running a genomics panel on patient sputum is far more costly (hundreds of dollars, depending on the method), can require a culture of the pathogen, which is both technically intensive and time consuming, and requires extensive bioinformatics assessment, all of which require skilled capabilities and complex laboratory infrastructure. Yet, whereas the Alere immunoassay interrogates for one single biomarker of interest, genomic arrays can provide a pan-diagnostic approach for discriminative diagnosis of infection. Similarly, proteomic and metabolomic arrays are time consuming, labor intensive and expensive. However, genomic technologies are advancing with respect to ease of use and operation at unprecedented rates, and in fact, have defied Moore’s law. Field forward sequencing capabilities that can be used quickly and in an automated version at the point of need are rapidly emerging. Many culture-free sequencing capabilities are emerging, decreasing the time to result extensively, especially when combined with deployable and easy to use informatics pipelines. Service centers providing proteomic microarray development and validation are rapidly emerging, and despite the research required and custom development involved, are now available for ~$100/sample. Thus, these methods which are largely more time consuming and expensive than conventional methods, hold more promise for the future, because of their holistic nature, flexibility and agility to being applied to a variety of human health challenges. Especially with pediatric TB, where the reliability of current diagnostics is very poor, a novel omics future does promise a better and more reliable future strategy. Currently, there are various researchers working on the development of deployable and easy to use informatics pipelines for proteomics, genomics, and other omics strategies.”
Reviewer 3 Report
Jakhar et al. discuss the impact of omics based approaches in developing Tuberculosis diagnosis approaches. The authors have discussed and presented the current state of Tb diagnosis and drawbacks very well. Here are a few minor comments:
1. It is interesting that the authors discuss briefly about non-coding RNAs. A recent study has explored few other small non-coding RNAs as biomarkers in Tb infection (DOI: 10.1128/mBio.01037-19) - it would be worth adding this study or other similar studies to the same discussion to give a complete picture of non-coding RNAs as biomarkers in Tb.
2. The authors have mentioned bioinformatic assessment of WGS not being easily deployed in resource poor setting as a potential drawback of WGS based diagnosis of Tb. However, this would also be the case for the other approaches like transcriptomics and proteomics as well - I think this drawback can be generalized for other approaches as well rather than for one specific omic approach - the discussion can be modified accordingly.
3. A few of the points seem to be repeated and redundant, the authors may minimize this to improve readability - for example - sputum-based diagnostics being difficult in pediatric population has been repeated several times in the manuscript.
Author Response
Manuscript ID # ijms-902902
Pediatric Tuberculosis: The Impact of “Omics” on Diagnostics Development
Authors’ Response to the Reviewers’ Comments:
We thank the reviewers for a meticulous assessment of our manuscript and making useful comments/remarks for further improvements. Our point-by-point responses to the reviewers’ comments are provided below.
Reviewer 3
- It is interesting that the authors discuss briefly about non-coding RNAs. A recent study has explored few other small non-coding RNAs as biomarkers in Tb infection (DOI:10.1128/mBio.01037-19) - it would be worth adding this study or other similar studies to the same discussion to give a complete picture of non-coding RNAs as biomarkers in Tb.
Thank you for this suggestion, we have included this and additional references on page 14 of the revised manuscript. This section now reads as follows: “Using RNA sequencing of whole blood samples, researchers examined the use of small noncoding RNA population such as miRNA, PIWI-interacting RNA (piRNA), small nucleolar RNA (snoRNA), and small nuclear RNA (snRNA) as host biomarkers for active and latent MTB in a systematic manner. From this approach, one miRNA and 2 piRNAs were identified as potential biomarkers for latent MTB, and further studies are required for their validation in such an application”.
- The authors have mentioned bioinformatic assessment of WGS not being easily deployed in resource poor setting as a potential drawback of WGS based diagnosis of Tb. However, this would also be the case for the other approaches like transcriptomics and proteomics as well - I think this drawback can be generalized for other approaches as well rather than for one specific omic approach - the discussion can be modified accordingly.
It is true that the availability of simple, ready to use bioinformatic strategies is a rate limiting step for the deployment of a variety of omics strategy, and this has been clarified in page 9 of the revised document as “The role of “omics” in TB diagnostic development” on page 9 and 10. Further, we have discussed informatics pipelines in development.
Further, on Page 10, we now note that “To date, there are various researchers working on the development of deployable and easy to use informatics pipelines for proteomics, genomics, and other omics strategies”
- A few of the points seem to be repeated and redundant, the authors may minimize this to improve readability - for example - sputum-based diagnostics being difficult in pediatric population has been repeated several times in the manuscript.
We have carefully combed through the manuscript and attempted to address redundancies, including the one mentioned above, and streamline the content more thoroughly.
Round 2
Reviewer 1 Report
The author has improved some of the suggestions I made in the current version. However, those following concerns have not been addressed.
1. There is a fair amount of proteomic studies reported, and it will be hard to summarize in two paragraphs. A proper expansion of the studies using proteomics would be required. For instance, in these studies, the host proteomics studies, the antigen proteomics studies, the protein microarray studies were not well described.
2. There are at least three kinds of Full Mtb protein array reported. The current review only mentioned one of them.
3. LAM would be of great value for diagnostics. A proper truncation and summary would be appreciated.
Author Response
Dear Reviewer
We have attempted to address your concerns as thoroughly as possible. In addition, we have edited the manuscript to ensure it is compliant with language requirements, grammatical structure and flow. We hope you will find these responses satisfactory, and the manuscript suitable for publication in the journal. Thank you for your consideration.
- There is a fair amount of proteomic studies reported, and it will be hard to summarize in two paragraphs. A proper expansion of the studies using proteomics would be required. For instance, in these studies, the host proteomics studies, the antigen proteomics studies, the protein microarray studies were not well described.
We have expanded on the proteomics section in order to address the reviewer’s comment. We have described host biomarker, MTB Pathogen and protein microarrays in the revised version of the manuscript. The section on page 15, lines 287 to page 19, line 361 now reads as follows:
“Proteomics profiling has been used to measure cellular activity, and can provide a deep insight into cellular processes in complex clinical backgrounds. Understanding the wide array of proteins expressed by both MTB and the host in response to MTB infection could shed light on pathways responsible for pathogenesis and persistence [83,84]. Such proteomic studies could target the pathogen-specific proteome, or host signatures in response to MTB infection, both of which have been attempted extensively, and examples from which are discussed below [85,86].
Early proteomic studies used two-dimensional gel electrophoresis (2D-GE) to analyze proteins from bacterial fractions and culture supernatants of MTB [59]. However, the low resolution of this method limited the analysis to only a few hundred proteins [87–90], which is insufficient to provide a clear assessment of the signature array. The use of liquid chromatography-tandem mass spectrometry (LC-MS/MS) shotgun proteomic methods in both targeted and non-targeted studies has allowed for the expansion of this capability to several thousand proteins at a given time [91,92]. More advanced MS techniques such as selected reaction monitoring has allowed for the quantification of ~80% of the MTB proteome, and does not require cell fractionation or separation [92]. In addition to MS, proteome microarrays have also been used to profile thousands of protein interactions in a single experiment [93,94]. Proteomic arrays have enabled researchers to define an immunoproteome for MTB. Until recently much of biomarker discovery has relied on traditional methods for separation and identification, however, as alternative methods are being constantly described the field has grown. To date there are three proteome-wide screening approaches that have been employed for the identification of candidate antigens for CD4+ T cell responses to MTB. All three studies found that a relatively small percentage of the proteome was responsible for the majority of the immune response [86,95,96].
In addition to an immunoproteome, a proteomic microarray approach has been used to screen 4,262 MTB antigens from 40 adult TB patients which allowed for the identification of 152 MTB antigens that were differentially elevated among patients with active versus latent disease [97]. Yet another study used a two-way proteome microarray approach to screen 84 potential host MTB interactors in infected adults, developing a signature repository that can be further used to understand MTB pathogenesis [98]. Deng et al. identified 14 adult serum biomarkers to differentiate between patients with active disease and those that have recovered from TB infection, facilitating monitoring of treatment outcomes [99]. In addition to presenting the proteome library, the investigators were also able to begin to explore the use of such microarrays in determining protein-protein interactions, biomarker discovery and differentiating between individuals with active disease and those that had recovered from TB infection, demonstrating the potential usefulness of such platforms for real-world applications.
From a longitudinal cohort of 6,363 MTB positive, HIV-negative adolescents of ages between 12-18 years in South Africa, host protein signatures associated with MTB were systematically assessed. In this study, the cohort was followed for 2 years and investigators reported that 46 individuals developed microbiologically confirmed MTB disease, while 106 non-progressors were identified. 3,000 human host proteins from plasma were quantified, of which 361 were found to demonstrate significant difference in abundance between individuals with microbiologically confirmed TB and non-progressors. From these 361 proteins, a 5-protein signature, TB Risk Model 5 (TRM5), was further sub-selected for use in discriminatory diagnostics. A second 3-protein pair (3PR) was further added to this sub-selection in order to improve the efficacy of the diagnostic platform. However, neither the TRM5 or 3PR achieved the minimum criteria for an incipient TB test as defined by FIND and WHO, and therefore, additional work is still needed to improve these signature-based protein assays [100]. Further studies such as this, and the concomitant development of sensitive diagnostic platforms can transform the landscape of empirical molecular diagnostics.
In a third study, researchers identified an eight-protein host signatures which had ramifications for the diagnosis of TB disease. In this study, three separate cohorts of were enrolled for a total of 640 individuals. The initial cohort of individuals was used for the screening of protein biomarkers of TB, the second to establish and test the predicted model, and the third for biomarker validation. The initial round of screening involved a microarray comprised of 16 non-overlapping arrays to measure 640 human proteins. Sixteen proteins of interest were then further analyzed in a second array. Using a series of mathematical models, a diagnostic model was built using an eight-protein signature. In the second test cohort the signature had an 83% specificity and a 76% sensitivity. The third cohort in which the signature was validated the specificity and sensitivity was 84% and 75% respectively. While this study was done with adults, a similar study could be designed for pediatric MTB to develop a pediatric specific model[101].
Proteomic profiles can allow for the diagnosis of pediatric TB, and despite the disease’s varied manifestations, several researchers have begun to specifically explore that possibility. For instance, a quantitative proteomics approach using LC-MS/MS was employed to characterize plasma from 72 children in different test groups (active TB, inflammatory disease control, healthy control) at a Beijing Children’s Hospital. The study identified 49 proteins in pediatric cases that were differentially expressed between active and latent TB [102]. One study characterized the plasma proteins in children at different MTB infection stages (active TB and LTBI), and identified four proteins -XRCC4, PCF11, SEMA4A, and ATP11A- to be signatures of active TB disease using proteomics [102]. Additionally, a subset of proteins that are exported, termed the exportome, could be potential source of additional disease biomarkers. Efforts to identify exported proteins have been typically limited to in vitro work. However, recently a in vivo method has been described and termed EXIT (exported in vivo technology) for the discovery of MTB exported proteins, as demonstrated in murine infection models. Over 500 proteins were revealed to be exported, several of which were induced in vivo. Proteins discovered by this technique should be further explored as potential biomarkers for adult and pediatric MTB disease markers[103]. Given the differential presentation of pediatric TB disease, it is likely that a combinatorial approach exploring varied biomarker signature profiles may provide a greater reliability of identification rather than a single factor approach [104].”
- There are at least three kinds of Full Mtb protein array reported. The current review only mentioned one of them.
We have revised the manuscript to include the three different types of the MTB proteomic studies that have been described. The use of two dimensional gel electrophoresis, shotgun proteomics and advanced MS-based methods to analyze and describe the MTB Proteome has now been described in some detail. It is noted that whereas these methods are being developed and have improved our understanding of TB pathophysiology, none of the array based techniques have been applied for pediatric disease diagnostics as yet. The section on Page 16, line 293 onwards now reads-
“The function of about one-quarter of the MTB coding genome and the precise activity and protein networks of most of the associated proteins remain poorly understood. Protein mass spectrometry and functional proteomics have provided new insights into making this information more accessible to diagnostics development. Early proteomic studies used two-dimensional gel electrophoresis (2D-GE) to analyze proteins from bacterial fractions and culture supernatants of MTB [59]. However, the low resolution of this method limited the analysis to only a few hundred proteins [87–90], which is insufficient to provide a clear assessment of the signature array. The use of liquid chromatography-tandem mass spectrometry (LC-MS/MS) shotgun proteomic methods in both targeted and non-targeted studies has allowed for the expansion of this capability to several thousand proteins at a given time [91,92]. More advanced MS techniques such as selected reaction monitoring has allowed for the quantification of ~80% of the MTB proteome, and does not require cell fractionation or separation [92]. In addition to MS, proteome microarrays have also been used to profile thousands of protein interactions in a single experiment [93,94]. Proteomic arrays have enabled researchers to define an immunoproteome for MTB. Until recently much of biomarker discovery has relied on traditional methods for separation and identification, however, as alternative methods are being constantly described the field has grown. To date there are three proteome-wide screening approaches that have been employed for the identification of candidate antigens for CD4+ T cell responses to MTB. All three studies found that a relatively small percentage of the proteome was responsible for the majority of the immune response [86,95,96].”
- LAM would be of great value for diagnostics. A proper truncation and summary would be appreciated.
We have attempted to greatly truncate this section and add critical summaries for this component. This section, line 407 onwards, reads
“LAM is an amphipathic molecule released from metabolically active or degrading bacterial cells resulting in the activation of host immune response [125,126][127]. In 2001, Hamasur et al. discovered that LAM was detectable in the urine several hours after intra-peritoneal injection of crude MTB cell wall extract into mice [128]. The observation provided researchers with an opportunity to evaluate LAM as a biomarker for the development of non-invasive [128,129] point-of-care tests for TB. As a result, a lateral flow urine LAM assay is currently available (Determine™ TB LAM Ag, Abbott Biotechnologies), with a sensitivity of 45% and specificity of 92% in HIV positive patients [130] and is recommended by the WHO only for use in HIV-positive adults with CD4 counts less than or equal to 100 cells·μL-1 presenting with symptoms of TB [38]. The guidelines for use of urine LF-LAM assay are similar in children, based on data from adults [38]. Previous work from our team demonstrated the detection of urinary LAM at a maximal concentration of 350 pM in individuals without HIV co-infection using a sandwich immunoassay on an ultra-sensitive waveguide based optical biosensor [42], which is supported by a new study using an improved chemiluminescence readout, with sensitivity and specificity of 93% and 97% respectively [47]. These findings show that a more sensitive assay format is required in immunocompetent individuals.
Despite advances in LAM diagnostics testing for adults, data on pediatric testing is still scarce. A WHO update on urine LAM assays reported a pooled sensitivity of 47%, and a pooled specificity of 82% among various studies performed in children with HIV [131], which is also reflected in independent assessments of both lateral flow and ELISA formats of detection [39][132][16]. These studies show that LAM measurement is more reliable in immunocompromised children, and that measured concentrations of the antigen decrease with anti-TB treatment [132], which suggests potential for this biomarker to be used as a prognostic indicator. Longitudinal studies demonstrating antigen concentrations as a function of disease progression and treatment must be performed in order to validate this hypothesis, as previously demonstrated for the measurement of lipomannan in M. bovis infection [133–136].
Previous work from our group has demonstrated that LAM is associated with High-density lipoproteins (HDL) in host blood [45]. This association should be considered when developing diagnostics, as with membrane insertion and lipoprotein capture methodologies [43,133,137–140], because traditional strategies for measuring the monomeric antigen are likely to be unsuccessful in this conformation. To date there are only a few studies showing detection of LAM in blood from adults [46,47,129] and none in a pediatric population.”